# Role of ivermectin in the prevention of SARS-CoV-2 infection among healthcare workers in India: A matched case-control study

Priyamadhaba Behera[1], Binod Kumar Patro[1], Arvind Kumar Singh[1], Pradnya Dilip Chandanshive[1], Ravikumar S. R.[1], Somen Kumar Pradhan[1], Siva Santosh Kumar Pentapati[1], Gitanjali Batmanabane[2]*, Prasanta Raghab Mohapatra[3], Biswa Mohan Padhy[4], Shakti Kumar Bal[3], Sudipta Ranjan Singh[5], Rashmi Ranjan Mohanty[6]

1 Department of Community Medicine and Family Medicine, All India Institute of Medical Sciences Bhubaneswar, Bhubaneswar, India, 2 All India Institute of Medical Sciences Bhubaneswar, Bhubaneswar, India, 3 Department of Pulmonary Medicine and Critical Care, All India Institute of Medical Sciences Bhubaneswar, Bhubaneswar, India, 4 Department of Pharmacology, All India Institute of Medical Sciences Bhubaneswar, Bhubaneswar, India, 5 Department of Forensic Medicine and Toxicology, All India Institute of Medical Sciences Bhubaneswar, Bhubaneswar, India, 6 Department of General Medicine, All India Institute of Medical Sciences Bhubaneswar, Bhubaneswar, India

* director@aiimsbhubaneswar.edu.in

**Data Availability Statement:** The anonymized data set has been uploaded to the public repository (figshare). The dataset can be accessed at the

## Abstract

### Background

Ivermectin is one among several potential drugs explored for its therapeutic and preventive role in SARS-CoV-2 infection. The study was aimed to explore the association between ivermectin prophylaxis and the development of SARS-CoV-2 infection among healthcare workers.

### Methods

A hospital-based matched case-control study was conducted among healthcare workers of AIIMS Bhubaneswar, India, from September to October 2020. Profession, gender, age and date of diagnosis were matched for 186 case-control pairs. Cases and controls were healthcare workers who tested positive and negative, respectively, for COVID-19 by RT-PCR. Exposure was defined as the intake of ivermectin and/or hydroxychloroquine and/or vitamin-C and/or other prophylaxis for COVID-19. Data collection and entry was done in Epicollect5, and analysis was performed using STATA version 13. Conditional logistic regression models were used to describe the associated factors for SARS-CoV-2 infection.

### Results

Ivermectin prophylaxis was taken by 76 controls and 41 cases. Two-dose ivermectin prophylaxis (AOR 0.27, 95% CI, 0.15–0.51) was associated with a 73% reduction of SARS-CoV-2 infection among healthcare workers for the following month. Those involved in physical activity (AOR 3.06 95% CI, 1.18–7.93) for more than an hour/day were more likely to contract SARS-CoV-2 infection. Type of household, COVID duty, single-dose ivermectin

following link: https://doi.org/10.6084/m9.figshare.13603454.

**Funding:** The author(s) have received no specific funding for this work.

**Competing interests:** The authors have declared that no competing interests exist.

prophylaxis, vitamin-C prophylaxis and hydroxychloroquine prophylaxis were not associated with SARS-CoV-2 infection.

## Conclusion

Two-dose ivermectin prophylaxis at a dose of 300 μg/kg with a gap of 72 hours was associated with a 73% reduction of SARS-CoV-2 infection among healthcare workers for the following month. Chemoprophylaxis has relevance in the containment of pandemic.

## Introduction

The SARS-CoV-2pandemic has claimed over 2,041,426 lives and affected over 95,567,167 persons worldwide by 18th January 2021 [1]. Meanwhile, the subcontinent of India has reported 10,572,672 active coronavirus disease 2019 (COVID-19) confirmed cases and 152,458 deaths related to the same virus [2]. The respiratory system is commonly involved in COVID-19, causing fever, cough and dyspnoea [3]. Alteration of the smell and taste, gastrointestinal symptoms, headache, and cutaneous manifestations are other associated symptoms that have been reported with COVID-19 [4–7].

Worldwide, healthcare workers (HCW) working on the frontline are at risk of exposure to the virus as they battle to save patients with COVID-19. A systematic review on infection and deaths in HCWs due to COVID-19 found that the number of infected HCWs workers ranged from 1,716 to 17,306, varying from country to country [8]. Another report in September 2020 stated that COVID-19 had infected nearly 570,000 HCWs while as many as 2,500 had succumbed to the disease in the region of the Americas [9]. Organizations at international and national levels have shared advisories and guidelines with measures to ensure the safety of HCWs while they serve amidst the COVID-19 pandemic. Provision of Personal Protective Equipment (PPE) while on duty, free-of-cost SARS-CoV-2 testing, timely payments, support helplines, online discussions, training, and capacity building for infection prevention and control are some of the measures taken to prevent the infection and spread of COVID-19 among HCWs [10–12].

For high-risk persons such as HCWs, additional safety measures are deemed necessary to prevent them from getting infected. During initial part of the pandemic, hydroxychloroquine (HCQ) as a chemoprophylaxis agent was used in India under the recommendation of a few experts with little evidence and lack of scientific data [13, 14]. The World Health Organization's(WHO) Solidarity trial's Executive Group and principal investigators decided to stop the hydroxychloroquine arm based on evidence from the Solidarity trial and UK's Recovery trial, which showed that HCQ did not result in the reduction of mortality of hospitalized COVID-19 patients compared with standard of care [15]. Ivermectin has been used as a therapeutic treatment of mild to moderate COVID-19 cases [16]. It has also been found to prevent symptoms of COVID-19 in post-exposure prophylaxis among HCWs [17]. A clinical trial conducted in Egypt in high-risk contacts of COVID-19 patients reported that compared to 7.4% of participants in the ivermectin arm, 58.5% in the control arm had symptoms suggesting SARS-CoV-2 infection [17]. A Few other studies have also shown favorable results with the use of ivermectin as prophylaxis and treatment [18, 19].

All India Institute of Medical Sciences, Bhubaneswar, is a tertiary care, government-funded teaching hospital situated in Odisha, in the Eastern part of India. From August 2020 onwards, a large number of HCWs employed in the hospital got infected, thus affecting hospital services.

Ivermectin has been shown to have diverse mechanisms by which it targets SARS-CoV-2and has a proven safety profile of many decades. These facts, along with the encouraging results of the study from Egypt, prompted us to explore the role of ivermectin as a prophylactic agent against COVID-19 in HCWs.

## Materials and methods

### Study population and sample size

The present study is a hospital-based matched case-control study conducted among HCWs of the All India Institute of Medical Sciences (AIIMS) in Bhubaneswar, Odisha, India, during September-October 2020. To calculate sample size, we assumed ivermectin prophylaxis in the control group to be 22% as there were no data available from prior studies. Considering, 80% power, 5% alpha, 1:1 matching of cases to controls, minimum discordant pairs to be detected was set to 54, with an expected odds ratio of 0.5, the sample size was estimated to be 186 pairs, i.e., 372 individuals [20]. Cases and controls were identified from the existing line list, which was prepared by the contact tracing team at AIIMS Bhubaneswar. The line list contained the list of the AIIMS Bhubaneswar HCWs' risk of exposure to SARS-CoV-2based on the assessment carried out using the WHO risk assessment guidelines [21]. This risk assessment helped in identifying population of cases and controls having similar risk. Cases were HCWs who were diagnosed as positive for COVID-19 by Reverse Transcription Polymerase Chain Reaction (RT-PCR). Controls were defined as HCWs who were diagnosed as negative for COVID-19 by RT-PCR with a similar risk of exposure toSARS-CoV-2. For every enrolled case, a control was selected from the existing line list. Individual matching was carried out for profession, gender and age. In addition, an attempt was made to match the date of diagnosis. However, when the match was not possible for the same date, the control was selected from the nearest possible date of diagnosis. In the majority of cases, it was within a week. The average number of days for a difference in date of diagnosis was 3.8 days between cases and controls. Exposure was defined as the prophylaxis viz., ivermectin and/or (HCQ) and/or vitamin C and/or other agent taken for the prevention of COVID-19. The HCWs of AIIMS Bhubaneswar were advised to take HCQ prophylaxis as per Indian Council of Medical Research (ICMR) guidelines from 11[th]April 2020 [22], in addition to using appropriate PPE depending on the place they were posted. However, the uptake was not encouraging on account of known cardiovascular adverse effects of HCQ. Further, on 17[th]September 2020, a decision to provide all HCWs with ivermectin for prophylactic use was announced, based on a consensus statement that was released at the institute (Box 1).

### Data collection and statistical analysis

The cases and controls dates range from 20th September 2020 to 19th October 2020. Participants were contacted for interviews from 19th October 2020 to 24th October 2020. After the selection of cases and controls, a phone call was made to each participant. The data related to COVID duty, family type, history of prophylaxis intake, history of hospital admission and physical activity were collected. (S1 Appendix) Data was entered in Epicollect5. Data cleaning and analysis was done using STATA version 13. The difference in characteristics of cases and controls were assessed using the chi-square test for categorical variables and t-test for continuous variables. Mean and standard deviation was used for continuous variables and proportion for categorical variables. Matched pair analysis was done using the McNemar chi-square test. Matched pair odds ratio was estimated for ivermectin, vitamin-C and HCQ prophylaxis. The potential confounders which could not be matched were adjusted during analysis with conditional logistic regression models. In model 1, we included the variables -COVID duty, family

## Box 1. AIIMS Bhubaneswar consensus statement for ivermectin prophylaxis among healthcare workers

Based on the long history of clinical use, favourable safety profile, and the reportedly promising effect of ivermectin as a prophylactic agent in COVID-19, the expert committee group proposes the following consensus statement:

Suggested prophylaxis for doctors/nurses/staff/students of AIIMS Bhubaneswar with ivermectin*

First Dose:

Ivermectin 300 µg/kg body weight on Day 1 (Directly Observed) and 4 (72 hours apart).

For 40–60 kg:15mg, 60–80 kg:18 mg, > 80 kg:24 mg

Subsequent dose: once a month dose (as above/kg body weight) on every 30th day after the last dose.

Ivermectin should be taken on an empty stomach with water.

*The above schedule will be followed till further guideline/new evidence is available.

*Pregnant women will not be given this drug. Women of childbearing age will be warned not to conceive while on this drug, in case they decide to take the drug.

This consensus statement had been prepared by a team of faculty members from various specialities and discussed and approved in the COVID-19 Working Group meeting on 11th September 2020.

type and physical activity (a proxy for social contacts), which may be the risk factors for COVID-19. In model 2, we included the variables- ivermectin, vitamin-C and HCQ, which were practiced as prophylaxis by HCWs for the prevention of COVID-19.

The study protocol was approved by the Institutional Ethics Committee of AIIMS, Bhubaneswar, via ref number: T/IM-NF/CM&FM/20/125. (S1 Protocol) Verbal informed consent was obtained telephonically before participation in the study. This consent procedure was approved by the ethics committee.

## Results

A total of 904 individuals working in AIIMS, Bhubaneswar, got tested for SARS-CoV-2 during one month (20th September 2020-19th October 2020). Out of 904 persons who were tested, 234 persons tested positive, and 670 persons tested negative for COVID-19. After matching with the profession, gender, age and date of diagnosis, there were 190 cases for which controls were available. Out of 190 cases, four did not give consent for participation. Therefore,186 matched pairs or 372 participants were finally included in our study. Participants had a mean (SD) age of 29± 6.83 years, and the mean difference in date of diagnosis between cases and control was 3.8 days. In one matched case-control pair, one intern was matched with a final year undergraduate student, which was the closest possible match for the case. The profession of the remaining 185 case-control pairs was perfectly matched (Table 1). Out of 186 cases, 18 (9.7%) cases were admitted to a hospital, while 168 (91.3%) cases opted for home isolation.

**Table 1. Characteristics of study participants(n = 372).**

|  | Control (n = 186) | Case (n = 186) | Total (n = 372) | P-value |
|---|---|---|---|---|
| **Mean age in years (Mean ± SD)** | 29.26 ± 6.61 | 29.25 ± 7.05 | 29.25 ± 6.83 | 0.99 |
| **Age groups** |  |  |  |  |
| <30 years | 110 (59.1%) | 116 (62.4%) | 226 (60.8%) | 0.73 |
| 30–39 years | 59 (31.7%) | 52 (28.0%) | 111 (29.8%) |  |
| 40–49 years | 15 (8.1%) | 14 (7.5%) | 29 (7.8%) |  |
| ≥50 years | 2 (1.1%) | 4 (2.1%) | 6 (1.6%) |  |
| **Gender** |  |  |  |  |
| Male | 125 (67.2%) | 125 (67.2%) | 250 (67.2%) | 1.00 |
| Female | 61 (32.8%) | 61 (32.8%) | 122 (32.8%) |  |
| **Whether you had COVID-19 duty[a] in the hospital?** |  |  |  |  |
| Yes | 107 (57.5%) | 106 (57.0%) | 213 (57.3%) | 0.92 |
| No | 79 (42.5%) | 80 (43.0%) | 159 (42.7%) |  |
| **Daily duration of physical activity** |  |  |  |  |
| No physical activity | 159 (85.5%) | 147 (79.0%) | 306 (82.3%) | 0.06 |
| Less than 30 minutes | 6 (3.2%) | 8 (4.3%) | 14 (3.7%) |  |
| 30–59 minutes | 14 (7.5%) | 11 (5.9%) | 25 (6.7%) |  |
| ≥60 minutes | 7 (3.8%) | 20 (10.8%) | 27 (7.3%) |  |
| **Profession** |  |  |  |  |
| Support staff | 52 (28.0%) | 52 (28.0%) | 104 (28.0%) | 1.00 |
| Residents | 36 (19.3%) | 36 (19.4%) | 72 (19.4%) |  |
| Nursing Officer | 64 (34.4%) | 64 (34.4%) | 128 (34.4%) |  |
| Intern | 12 (6.5%) | 13 (7.0%) | 25 (6.7%) |  |
| Student | 13 (7.0%) | 12 (6.4%) | 25 (6.7%) |  |
| Faculty | 9 (4.8%) | 9 (4.8%) | 18 (4.8%) |  |
| **Type of household** |  |  |  |  |
| Extended family | 86 (46.2%) | 98 (52.7%) | 183 (49.5%) | 0.26 |
| Nuclear family | 28 (15.1%) | 20 (10.7%) | 48 (12.9%) |  |
| With friend or roommate | 64 (34.4%) | 55 (29.6%) | 119 (32.0%) |  |
| Living alone | 8 (4.3%) | 13 (7.0%) | 21 (5.6%) |  |
| Total | 186 (100.0%) | 186 (100.0%) | 372 (100.0%) |  |

[a]COVID-19 duty was defined as HCWs involved in COVID-19 patient care in the outpatient department (OPDs) and/or inpatient department (IPDs) and/or intensive care unit (ICU).

The majority of the participants (60.75%) were below 30 years of age. Nearly two-thirds of participants (67.2%) were male. More than half of the participants (57.26%) were involved in COVID-19 patient care in the outpatient department (OPDs) and/or inpatient department (IPDs) and/or intensive care unit (ICU) in the last one month. Most participants (82.26%) were not doing any physical activity during the study period. Among the various modes of physical activity, jogging and yoga were chosen by 38 participants each (10.22%), gymnasium by 15 (4.03%), and sports by 11 (2.96%) participants. Among 372 participants, 128 (34.41%) were nursing officers, 104 (27.96%) were supporting staffs, 72 (19.35%) were resident doctors, 25 (6.72%) were interns, 25 (6.72%) were students and 18 (4.84%) were faculty members. Half of the participants (49.33%) were staying in an extended family, one-third of participants (32.08%) were staying with friends, and others were either staying in a nuclear family (12.94%) or living alone (5.66%) (Table 1).

Out of 372 participants, 171 participants (101 from cases and 70 from controls) took any form of prophylaxis. Hundred and seventeen (31.4%) participants had a history of ivermectin prophylaxis-76 from controls and 41 from cases, 67 (18.01%) participants had a history of vitamin-C prophylaxis-38 from controls and 29 from cases, 19(5.11%) participants had a history of HCQ prophylaxis-12 from controls and seven from cases (Table 2). Four participants were taking home-based remedies for the prevention of COVID-19. Ninety-one (24.46%) participants had a history of two-dose ivermectin prophylaxis (300 µg/kg at Day 1 and Day 4). However, 17 (4.57%) participants took only one dose (300 µg/kg), and 9 (2.42%) participants continued the same dose for three or more days. Out of 67 participants, who took vitamin-C prophylaxis, 54 participants took a dose of 500 mg once daily, and 13 participants took vitamin-c 500 mg twice daily. The majority of participants took vitamin-C for less than one month; however, 27 participants were continuing vitamin-C prophylaxis for more than one month. HCQ prophylaxis was practiced 400 mg once a week. Out of 19 participants who took HCQ prophylaxis, ten participants took for three or more weeks, five participants took for two weeks, and four participants took for a week.

In the matched pair analysis, ivermectin prophylaxis (OR 0.35, 95% CI, 0.20–0.60) was associated with the reduction of SARS-CoV-2 infection. However, vitamin-C prophylaxis (OR 0.71, 95% CI, 0.40–1.26) and HCQ prophylaxis (OR 0.58, 95% CI, 0.19–1.61) had no significant association with SARS-CoV-2 infection (Table 3). In the multivariate conditional logistic regression model 2, two-dose of ivermectin prophylaxis (AOR 0.27, 95% CI, 0.15–0.51) was associated with a reduction of SARS-CoV-2 infection after adjusting for COVID duties, type of household, physical activity, vitamin-C prophylaxis and HCQ prophylaxis. However, physical activity for more than one hour was an independent risk factor (AOR 3.06 95% CI, 1.18–7.93) for SARS-CoV-2 infection (Table 4).

## Discussion

Our study has shown that two doses of ivermectin prophylaxis at a dose of 300 µg/kg given 72 hours apart was associated with a 73% reduction of SARS-CoV-2 infection among HCWs for the following month. Our results are similar to the randomized trial conducted by Waheed Shouman from the Zagazig University of Egypt. Out of the 203HCWs in the intervention arm,

Table 2. Comparison of nature of prophylaxis between the cases (n = 186) and controls (n = 186).

| Variable | Control N (%) | Case N (%) | Total N (%) |
|---|---|---|---|
| **History of any prophylaxis** | | | |
| No | 85 (45.7) | 116 (62.37) | 201 (54.0) |
| Yes | 101 (54.3) | 70 (37.6) | 171 (46.0) |
| **History of intake of ivermectin for COVID-19 prophylaxis** | | | |
| No | 110 (59.1) | 145 (78.0) | 255 (68.6) |
| Yes | 76 (40.9) | 41 (22.0) | 117 (31.4) |
| **History of intake of vitamin-C for COVID-19 prophylaxis** | | | |
| No | 148 (79.6) | 157 (84.4) | 305 (82.0) |
| Yes | 38 (20.4) | 29 (15.6) | 67 (18.0) |
| **History of intake of HCQ[a] for COVID-19 prophylaxis** | | | |
| No | 174 (93.5) | 179 (96.2) | 353 (94.9) |
| Yes | 12 (6.5) | 7 (3.8) | 19 (5.1) |
| Total | 186 (100.00) | 186 (100.00) | 372 (100.00) |

[a]Hydrochloroquone (HCQ).

**Table 3. Matched pair analysis of exposure/prophylaxis taken for COVID-19 (n = 186).**

| | Variables | Controls | | Total | McNemar's Chi-square |
|---|---|---|---|---|---|
| | | Ivermectin | No ivermectin | | |
| Cases | **Ivermectin** | 22 (29.0%) | 19 (17.3%) | 41 (22.0%) | χ2 = 16.78 |
| | **No ivermectin** | 54 (71.0%) | 91 (82.7%) | 145 (78.0%) | P<0.001 |
| | Total | 76 (100.0%) | 110 (100.0%) | 186 (100.0%) | |
| | Matched pair OR[a] | **0.35 (95% CI, 0.20 to 0.60)** | | | |
| | | Vitamin-C | No vitamin-C | | |
| | **Vitamin-C** | 6 (15.8%) | 23 (15.5%) | 29 (15.6%) | χ2 = 1.47 |
| | **No vitamin-C** | 32 (84.2%) | 125 (84.5%) | 157 (84.4%) | P = 0.22 |
| | Total | 38 (100.0%) | 148 (100.0%) | 186 (100.0%) | |
| | Matched pair OR | 0.71 (95% CI, 0.40 to 1.26) | | | |
| | | HCQ | No HCQ | | |
| | **HCQ[b]** | 0 (0.0%) | 7 (4.0%) | 7 (3.8%) | χ2 = 1.32 |
| | **No HCQ[b]** | 12 (100.0%) | 167 (96.0%) | 179 (96.2%) | P = 0.25 |
| | Total | 12 (100.0%) | 174 (100.0%) | 186 (100.0%) | |
| | Matched pair OR | 0.58 (95% CI, 0.19 to 1.61) | | | |

[a]Oddss Ratio (OR) and
[b]Hydroxychloroqune (HCQ).

**Table 4. Conditional logistic regression models for associated factors of SARS-CoV-2 infection.**

| Variable | Unadjusted Odds ratio (95% CI) | p-value | Model-1 | | Model-2 | |
|---|---|---|---|---|---|---|
| | | | Adjusted Odds ratio (95% CI) | p-value | Adjusted Odds ratio (95% CI) | p-value |
| **History of COVID-19 duty** | | | | | | |
| No | Reference | | Reference | | | |
| Yes | 0.96 (0.55 to1.66) | 0.89 | 0.93(0.51 to 1.67) | 0.81 | 0.88 (0.45 to 1.68) | 0.69 |
| **Household type** | | | | | | |
| Extended family | Reference | | Reference | | | |
| Friend/roommate | 0.68 (0.39 to 1.18) | 0.18 | 0.69(0.39 to 1.21) | 0.19 | 0.80 (0.43 to 1.48) | 0.48 |
| Nuclear family | 0.61(0.31 to 1.19) | 0.15 | 0.64(0.32 to 1.27) | 0.20 | 0.66 (0.32 to 1.35) | 0.25 |
| Alone | 1.51 (0.49 to 4.66) | 0.47 | 3.25(0.53 to 5.45) | 0.37 | 1.43 (0.42 to 4.91) | 0.57 |
| **Physical activity** | | | | | | |
| No physical activity | Reference | | Reference | | | |
| Less than 30 minutes per day | 1.33(0.46 to 3.84) | 0.59 | 1.34 (0.46 to 3.92) | 0.60 | 1.14 (0.36 to 3.61) | 0.82 |
| 30–59 minutes per day | 0.80(0.35 to 1.83) | 0.60 | 0.76 (0.33 to 1.79) | 0.53 | 0.61 (0.24 to 1.57) | 0.31 |
| ≥60 minutes per day | **2.83 (1.20 to 6.71)** | **0.02** | **2.86 (1.19 to 6.87)** | **0.02** | **3.06 (1.18 to 7.93)** | **0.02** |
| **Ivermectin prophylaxis** | | | | | | |
| No Ivermectin prophylaxis | Reference | | | | Reference | |
| Single-dose Ivermectin prophylaxis | 1.23(0.43 to 3.50) | 0.70 | | | 1.30 (0.44 to 3.85) | 0.63 |
| Two or more doses Ivermectin prophylaxis | **0.27(0.14 to 0.47)** | **0.00** | | | **0.27 (0.15 to 0.51)** | **0.00** |
| **Vitamin-C prophylaxis** | | | | | | |
| No | Reference | | | | Reference | |
| Yes | 0.72(0.42 to 1.27) | 0.23 | | | 0.82 (0.45 to 1.57) | 0.58 |
| **Hydroxychloroquine prophylaxis** | | | | | | |
| No | Reference | | | | Reference | |
| Yes | 0.58(0.23 to 1.48) | 0.26 | | | 0.56 (0.19 to 1.63) | 0.29 |

only 7.4% developed symptoms versus 58.5% of the 101 HCWs in the control arm after 14 days of enrolment. The study also reported no mortality or serious adverse events due to ivermectin in the intervention arm [17]. Mostly, ivermectin prophylaxis will benefit the HCWs who are vulnerable to the infection because of their profession. Our study findings throw light in the same direction that ivermectin may play a vital role in the prevention strategy of SARS-CoV-2 infection.

Our study also documented that single-dose ivermectin prophylaxis at a dose of 300 μg/kg, HCQ prophylaxis, and vitamin-C prophylaxis are not associated with preventing SARS-CoV-2 infection. Engaging in physical activity for more than one hour daily, which is taken for lack of physical distancing, was an independent risk factor for SARS-CoV-2 infection in our study. Study participants practiced outdoor physical activity like walking or jogging, many also worked out at the gymnasium, and a few were involved in playing sports. The possible explanations may be that physical activity poses a greater risk of exposure to infection due to increased chances of social contact, difficulty wearing masks, and sharing of gymnasium equipment by multiple persons, thereby rendering individuals vulnerable to infection. Various studies have emphasized the need and effectiveness of social or physical distancing as a preventive measure against COVID-19 while maintaining the importance of hygiene measures, use of face masks, and increase in testing facilities to prevent and reduce COVID-19 transmission [23–27].

The study by Caly et al. on the effect of ivermectin acting in vitro on Vero-hsLAM cells infected with COVID-19 has been quoted widely for providing the basis of usage of Ivermectin in COVID-19 [28]. It was found that the Vero/hsLAM cells infected with SARS-CoV-2 isolate Australia/VIC01/2020 and treated with 5μM ivermectin showed a 93% reduction in the viral RNA compared to the vehicle DMSO (Dimethyl sulfoxide) at the end of 24 hours duration. As much as 5000-fold reduction of viral RNA was observed at the end of 48 hours duration in the ivermectin-treated samples compared to the control samples. This observation indicated that approximately all viral material was eradicated by ivermectin treatment in 48 hours duration. However, at 72 hour-duration, there was no further reduction observed in the viral RNA levels. The study also determined the IC50 (half maximal inhibitory concentration) of ivermectin treatment to be 2.5μM under the test conditions [28].

Our study also estimated that single-dose prophylaxis has no association with a reduction of SARS-CoV-2 infection, and two-dose of ivermectin (300 μg/kg) was associated with a reduction of SARS-CoV-2 infection. Our study finding is supported by another study conducted by Chang et al., which found ivermectin to be useful as prophylaxis among healthcare personnel [18]. Their study aimed to investigate specifically post-exposure prophylaxis in contacts who tested negative for SARS-CoV-2 with ivermectin dose of 0.2mg/kg body weight on day one with an additional second dose of ivermectin on day 2 or 3 for men aged more than 45 years [18]. In contrast, our study investigated the healthcare workers who had tested for COVID-19 with positive cases and negative controls. We also matched the cases and controls according to age, gender, designation, and date of testing. Also, our sample size was larger than the former study.

A randomized controlled study conducted by Boulware et al. tested hydroxychloroquine as postexposure prophylaxis among high-risk contacts of confirmed COVID-19 cases. Similar to our study findings, their study reported no significant difference between the HCQ arm and the placebo arm. There were also more side effects reported with hydroxychloroquine in their study [29].

In literature, the proposed hypothesis is that vitamin C may have a role in preventing SARS-CoV-2 infection due to its potent antioxidant and immunomodulatory effects [30–32].

However, no research or hardcore evidence supports these findings. In our study, we did not find any association between vitamin-C prophylaxis and prevention of SARS-CoV-2 infection.

The strengths of our study are the adequate sample size, completeness of the data collection and verification from subjects. All the HCWs received ivermectin procured from a single manufacturer and belonged to the same batch for each strength. We adjusted for confounders by matching and multivariate analysis. Though recall bias is inherent in case-control studies, our data regarding the drug intake within the last one month is less likely to be forgotten by HCWs. Due to its observational nature, our study's findings need further confirmation using longitudinal studies or interventional studies to strengthen the evidence before its large-scale use among HCWs and the implementation of public health programs.

## Conclusion

We conclude that two-dose ivermectin prophylaxis at a dose of 300 μg/kg body weight with a gap of 72 hours was associated with a 73% reduction of SARS-CoV-2 infection among HCWs in the following one month. Chemoprophylaxis has relevance in the containment of pandemic. This is an intervention worth replicating at other centers until a vaccine is widely available.

## Supporting information

**S1 Appendix. Questionnaire used in the study (English and Odia language).**
(PDF)

**S1 Protocol. Study protocol.**
(DOC)

## Acknowledgments

We thank Mrs. Jyoti Muthusami for her assistance with language editing that significantly improved the manuscript. We are thankful to all the participants for their involvement in the study.

## Author Contributions

**Conceptualization:** Priyamadhaba Behera, Binod Kumar Patro, Gitanjali Batmanabane, Prasanta Raghab Mohapatra, Biswa Mohan Padhy, Shakti Kumar Bal, Rashmi Ranjan Mohanty.

**Data curation:** Pradnya Dilip Chandanshive, Ravikumar S. R., Somen Kumar Pradhan, Siva Santosh Kumar Pentapati.

**Formal analysis:** Priyamadhaba Behera, Pradnya Dilip Chandanshive, Ravikumar S. R.

**Investigation:** Priyamadhaba Behera, Binod Kumar Patro, Somen Kumar Pradhan, Gitanjali Batmanabane.

**Methodology:** Priyamadhaba Behera, Binod Kumar Patro, Arvind Kumar Singh, Ravikumar S. R., Somen Kumar Pradhan, Siva Santosh Kumar Pentapati, Gitanjali Batmanabane.

**Project administration:** Priyamadhaba Behera, Binod Kumar Patro, Arvind Kumar Singh, Gitanjali Batmanabane.

**Resources:** Priyamadhaba Behera, Arvind Kumar Singh, Gitanjali Batmanabane, Sudipta Ranjan Singh.

**Supervision:** Priyamadhaba Behera, Binod Kumar Patro, Arvind Kumar Singh, Gitanjali Batmanabane.

**Writing – original draft:** Priyamadhaba Behera, Pradnya Dilip Chandanshive, Gitanjali Batmanabane.

**Writing – review & editing:** Priyamadhaba Behera, Binod Kumar Patro, Arvind Kumar Singh, Somen Kumar Pradhan, Siva Santosh Kumar Pentapati, Gitanjali Batmanabane, Prasanta Raghab Mohapatra, Biswa Mohan Padhy, Shakti Kumar Bal, Sudipta Ranjan Singh, Rashmi Ranjan Mohanty.

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
