## [Decision Letter · Decision Letter 0]

15 Jan 2021

PONE-D-20-38031

Role of ivermectin in the prevention of COVID-19 infection among healthcare workers in India: A matched case-control study

PLOS ONE

Dear Dr. Batmanabane,

Thank you for submitting your manuscript to PLOS ONE. After careful consideration, we feel that it has merit but does not fully meet PLOS ONE’s publication criteria as it currently stands. Therefore, we invite you to submit a revised version of the manuscript that addresses the points raised during the review process.

Please see below comments made by the reviewers and provide a point by point response in your revised manuscript. Manuscript also need a professional English language editing prior to resubmission.

We look forward to receiving your revised manuscript.

Kind regards,

Muhammad Adrish

Academic Editor

PLOS ONE

Journal Requirements:

2. In the Methods section, please clearly state the date range over which cases and controls were identifying for the study using retrospective chart review. And please specify the date range over which participants were contacted for interview.

5. Please include captions for your Supporting Information files at the end of your manuscript, and update any in-text citations to match accordingly. Please see our Supporting Information guidelines for more information: http://journals.plos.org/plosone/s/supporting-information

Reviewers' comments:

Reviewer's Responses to Questions

**Comments to the Author**

1. Is the manuscript technically sound, and do the data support the conclusions?

Reviewer #1: Partly

Reviewer #2: Yes

2. Has the statistical analysis been performed appropriately and rigorously? 

Reviewer #1: Yes

Reviewer #2: Yes

3. Have the authors made all data underlying the findings in their manuscript fully available?

Reviewer #1: Yes

Reviewer #2: Yes

4. Is the manuscript presented in an intelligible fashion and written in standard English?

Reviewer #1: Yes

Reviewer #2: Yes

5. Review Comments to the Author

Reviewer #1: Priyamadhaba Behera et al. conducted an interesting study about the role of ivermectin in SARS-CoV-2 prophylaxis.

General comment

In the manuscript, the authors wrote "COVID-19 infection" (title included), but it is a disease. Please modify it with "SARS-CoV-2 infection".

Abbreviations should be written entirely in the first apparition in the text (e.g., COVID-19). Please, recheck the manuscript and fix it.

Some sentences are hard to follow. Some typos are present. I suggest re-reading the manuscript and fix it.

Introduction

The authors wrote, "The SARS-CoV-2 pandemic has claimed over 1,101,298 lives and affected over 39,196,259 persons worldwide". While I am writing the revision, the number of infected persons is 88,024,536, and deaths are 1,899,015. Please update these numbers, adding the date.

I believe it is important for the reader to write a sentence to present COVID-19 disease, particularly clinical presentation, describing both major (fever, cough, and dyspnea) and minor symptoms (alteration of the smell and taste, gastrointestinal symptoms, headache, and cutaneous manifestations). You could read and use these articles: https://doi.org/10.1002/hed.26269, https://doi.org/10.1002/hed.26204, https://doi.org/10.26355/eurrev_202007_22291, https://doi.org/10.1016/S1473-3099(20)30402-3, https://doi.org/10.1097/IPC.0000000000000952
https://doi.org/10.1111/eci.13427.

Methods

I believe that an important confounder is the typology of wards where HCWs were working. For example, people who work in intensive care units have an increased risk that people who work in low-intensity wards are plausible that the first are exposed to high SARS-CoV-2 viral load during some procedure (e.g., Tracheal intubation, Broncho-Alveolar Lavage). On the contrary, I do not believe that the profession has a role in infection risk. Please comment.

Furthermore, I would like to know if all participants used the same personal protective equipment, or maybe some wards used different PPE.

Besides, I would like to know if each HCW were free to choose to start a prophylactic treatment and which treatment. Another bias could be that people who decided to start a prophylactic treatment were more careful during the hospital's permanence and everyday life.

Result

Have the authors analysed if positive people who took ivermectin had a mild disease than positive people who have not taken ivermectin?

I suggest specifying that the number between brackets are odd ratios.

Tables

In all tables, the authors should add an explanation of the abbreviation used.

Table 1. It is not clear the meaning of "Whether you had COVID-19 duties in the hospital?". It is not clear if the "Duration of physical activity" is daily.

Reviewer #2: The paper entitled "Role of ivermectin in the prevention of COVID-19 infection among healthcare workers in India: A matched case-control study".The study was aimed to explore the association between ivermectin prophylaxis and development of COVID-19 infection among healthcare workers.

I suggest minor revisions in particular its important detail Table 3 because it is not clear; the conclusion have to be revised and clarify the bibliography "24".

6. PLOS authors have the option to publish the peer review history of their article (what does this mean?). If published, this will include your full peer review and any attached files.

Reviewer #1: No

Reviewer #2: No

---

## [Author Response · Author response to Decision Letter 0]

24 Jan 2021

Ref: PONE-D-20-38031

Manuscript title: Role of ivermectin in the prevention of SARS-CoV-2 infection among healthcare workers in India: A matched case-control study

We sincerely thank the editor and reviewer for their helpful suggestions and comments. We have revised the manuscript as per the suggestions and comments of reviewers. The revised manuscript has improved in content and presentation. A point to point response of each of the reviewer’s comment is given below. The revised manuscript (track file and clean file) along with reviewer comments is attached in the revised submission. 

Journal Requirements: 

Editor comment 1: 

 Author Response: We have formatted the manuscript as per PLOS ONE's style requirements. Thank you. 

Editor comment 2: 

2. In the Methods section, please clearly state the date range over which cases and controls were identifying for the study using retrospective chart review. And please specify the date range over which participants were contacted for interview.

Author Response: The cases and controls dates range from 20th September 2020 to 19th October 2020. Participants were contacted for interviews from 19th October 2020 to 24th October 2020. This has been added to the method section of manuscript. Thank you. Page no: 6, Line no: 122-123

Editor comment 3: 

 Author Response: The questionnaire (both language) used in the study has been uploaded as supporting information (S1 Appendix). Thank you. 

Editor comment 4: 

Author Response: The anonymized data set has been uploaded to the public repository (figshare). The dataset can be accessed at the following link -https://doi.org/10.6084/m9.figshare.13603454. 

Editor comment 5: 

5. Please include captions for your Supporting Information files at the end of your manuscript, and update any in-text citations to match accordingly. Please see our Supporting Information guidelines for more information: http://journals.plos.org/plosone/s/supporting-information

Author Response:

The captions for the supporting information files (S1 Appendix and S2 Protocol) are added at the end of the manuscript and in-text citations are updated to match accordingly. 

Supporting Information files:

S1 Appendix. Questionnaire used in the study (English and Odia language)

S2 Protocol. Study protocol.

Review Comments to the Author

Reviewer #1:

Reviewer comment:

Priyamadhaba Behera et al. conducted an interesting study about the role of ivermectin in SARS-CoV-2 prophylaxis.

Author Response: Thanks for your compliment. We have tried our best to address all your queries. 

General comment:

Reviewer comment:

 In the manuscript, the authors wrote "COVID-19 infection" (title included), but it is a disease. Please modify it with "SARSCoV-2 infection".

Author Response: COVID-19 infection has been replaced with “SARSCoV-2 infection” in the title and other places of manuscript. Thanks for your valuable suggestion. 

Reviewer comment:

Abbreviations should be written entirely in the first apparition in the text (e.g., COVID-19). Please, recheck the manuscript and fix it.

Author Response: The suggestion has been incorporated and the manuscript has been revised accordingly. Thank you. 

Page No: 3,4,5

Line No: 51, 56, 63-64, 69-70, 71-72, 94, 104-105, 114

Reviewer comment:

Some sentences are hard to follow. Some typos are present. I suggest re-reading the manuscript and fix it. 

Author Response: 

Thank you for identifying these essential typos and errors. The manuscript has been edited extensively for language. Senior authors have reread the manuscript, and the necessary corrections were made. We edited language with a professional's help, and her contribution has been acknowledged in the acknowledgment section.

 Introduction 

Reviewer comment:

The authors wrote, "The SARS-CoV-2 pandemic has claimed over 1,101,298 lives and affected over 39,196,259 persons worldwide". While I am writing the revision, the number of infected persons is 88,024,536, and deaths are 1,899,015. Please update these numbers, adding the date. 

Author Response: The numbers of SARS-CoV-2 infections and deaths have been updated till 18th January 2021. and the introduction of the manuscript has been modified accordingly. Page No: 3, Line No: 49-52

Reviewer comment:

I believe it is important for the reader to write a sentence to present COVID-19 disease, particularly clinical presentation, describing both major (fever, cough, and dyspnea) and minor symptoms (alteration of the smell and taste, gastrointestinal symptoms, headache, and cutaneous manifestations). You could read and use these articles: https://doi.org/10.1002/hed.26269, https://doi.org/10.1002/hed.26204, https://doi.org/10.26355/eurrev_202007_22291, https://doi.org/10.1016/S1473-3099(20)30402-3, https://doi.org/10.1097/IPC.0000000000000952

Author Response: We have incorporated the reviewer suggestions and revised the manuscript accordingly. Page No: 3, Line No: 52-55

Methods

Reviewer comment:

I believe that an important confounder is the typology of wards where HCWs were working. For example, people who work in intensive care units have an increased risk that people who work in low-intensity wards are plausible that the first are exposed to high SARS-CoV-2 viral load during some procedure (e.g., Tracheal intubation, Broncho-Alveolar Lavage). On the contrary, I do not believe that the profession has a role in infection risk. Please comment. 

Author Response: 

Supportive staffs, nursing officers, resident doctors, interns, students and faculties (senior doctors) were our participants. Students were not involved in COVID-19 patient care. Supportive staff, nursing officers, resident doctors, interns, and faculties (senior doctors) had different roles leading to differential exposure to the virus during COVID-19 patient care. As per administrative decision there are rotation of duties area among most of the Healthcare Workers in COVID ward and ICU. After one-week duty in COVID area, there occurs compulsory break of duty (off) under observation and followed by duties in other non-COVID areas of the hospitals. 

Hence, we thought adjusting for the profession during matching will minimize the selection bias. We agree with your suggestion that HCWs in intensive care units have an increased risk that people who work in low-intensity wards. During the pandemic, the staff in our hospital involved in COVID-19 patient care were rotated in outpatient department (OPDs), inpatient department (IPDs) and intensive care unit (ICU) during service delivery. Hence, the staff who had COVID-19 duties may have similar exposure. Therefore, we have considered COVID-19 duty as a variable for our analysis purpose. Thank you. 

Reviewer comment:

Furthermore, I would like to know if all participants used the same personal protective equipment, or maybe some wards used different PPE. Besides, I would like to know if each HCW were free to choose to start a prophylactic treatment and which treatment. 

Author Response:

Personal protective equipment (PPE) used during the duties were equal and compulsory for all participants. Yes, HCW were free to choose to start a prophylactic treatment. Ivermectin was made available free of cost to the HCWs. Uptake of ivermectin prophylaxis was voluntary. As per consensus statement prepared by institute (Panel 1), for ivermectin prophylaxis among healthcare workers- two-dose ivermectin prophylaxis at a dose of 300 μg/kg body weight with a gap of 72 hours was suggested. Majority participants took the suggested ivermectin prophylaxis, however, 17 participants did not comply to the second dose of ivermectin and nine participants preferred to continue the same dose for three or more days. Thank you. 

Reviewer comment:

Another bias could be that people who decided to start a prophylactic treatment were more careful during the hospital's permanence and everyday life. 

Author Response:

Ivermectin was offered to each HCW irrespective of exposure. Our hospital building is a single structure and central air-conditioning system. We had a strong institutional policy in place related to COVID-19 appropriate behavior (CAB) in the workplace, which may have avoided the possible bias. There was more stringent action by state authority for the same to control the outdoor transmissions.

 Thank you.

Result 

Reviewer comment:

Have the authors analysed if positive people who took ivermectin had a mild disease than positive people who have not taken ivermectin? 

Author Response: Thanks for raising an important research question. We were also interested in looking into the same in our data. Out of 186 cases, only 18 (9.7%) cases were admitted to a hospital, and the number of patients with severe COVID-19 disease was significantly less. Hence, our study was not powered for the suggested subgroup analysis. This is one of our research questions which we were looking to answer in our further studies.

Reviewer comment:

I suggest specifying that the number between brackets are odd ratios. 

Author Response: 

We have mentioned Odd Ratio (OR) and Adjusted Odd Ratio (AOR) in the number between brackets as suggested. Thank you. Page No: 2 and 8, Line No: 38, 40, 186-194

Tables 

Reviewer comment:

In all tables, the authors should add an explanation of the abbreviation used. 

Author Response: In all tables, the abbreviations have been explained. Thank you. Page No. 19-21 Line No. 448-449, 452 and 455. 

Reviewer comment:

Table 1. It is not clear the meaning of "Whether you had COVID-19 duties in the hospital?".

 It is not clear if the "Duration of physical activity" is daily.

Author Response:

COVID-19 duty was defined as HCWs involved in COVID-19 patient care in the outpatient department (OPDs) and/or inpatient department (IPDs) and/or intensive care unit (ICU). The data for the daily duration of physical activity was captured. We have incorporated these clarities in result section as well as Table 1. Thank you. Page No: 7 and 18-19 Line No: 158-160 and 448-449. 

Reviewer #2: 

Reviewer comment:

The paper entitled "Role of ivermectin in the prevention of COVID-19 infection among healthcare workers in India: A matched case-control study". The study was aimed to explore the association between ivermectin prophylaxis and development of COVID-19 infection among healthcare workers. I suggest minor revisions in particular its important detail Table 3 because it is not clear.

Author Response:

This study was designed to answer the following research questions – (Supporting Information file-S2 Protocol)

1. Whether oral ivermectin can be used for the prevention of COVID-19 disease?

2. If so, what is the probable dose of ivermectin which will benefit maximum in the Indian context?

Table 3-McNemar’s Chi-square test for matched-pair analysis answers the first research question. 

Ivermectin prophylaxis (OR 0.35, 95% CI, 0.20-0.60) was associated with the reduction of SARS-CoV-2 infection. However, vitamin-C prophylaxis (OR 0.71, 95% CI, 0.40-1.26) and HCQ prophylaxis (OR 0.58, 95% CI, 0.19-1.61) had no significant association with SARS-CoV-2 infection. (Table 3) 

Multivariate conditional logistic regression models help us adjust for potential confounders that we cannot adjust during matching and find the answer for the second research question. Two-dose ivermectin prophylaxis (0.27, 95% CI, 0.15-0.51) was associated with reducing SARS-CoV-2 infections. However, there was no association with single-dose ivermectin prophylaxis (1.30, 95% CI 0.44-3.85). (Table 4). 

We have provided more clarity about this in the result section of the manuscript. Thank you. 

Page No: 8, Line No: 186-194.

Reviewer comment:

The conclusion has to be revised 

Author Response: 

The conclusion has been modified as per your valuable suggestion. Thank you. 

 Page no: 11, Line no: 263-264

Reviewer comment:

Clarify the bibliography "24".

Author Response:

Bibliography "24" was not correctly placed. Thank you for reading the manuscript carefully and identifying the important error. The references have been updated and the manuscript has been revised. 

Corrected Reference:

Reference 18: Aguirre Chang G, Trujillo Figueredo A. COVID-19: ivermectin prophylaxis in adult contacts. First Report on Health Personnel and Post-Exposure Prophylaxis. Research Gate [Internet]. 2020 Jul 21 [cited 2020 Nov 6] Available from: https://www.researchgate.net/publication/344251319_COVID-19_ivermectin_prophylaxis_in_adult_contacts_First_Report_on_Health_Personnel_and_Post-Exposure_Prophylaxis.

Page No: 10, Line No: 235.

---

## [Decision Letter · Decision Letter 1]

3 Feb 2021

Role of ivermectin in the prevention of SARS-CoV-2 infection among healthcare workers in India: A matched case-control study

PONE-D-20-38031R1

Dear Dr. Batmanabane,

We’re pleased to inform you that your manuscript has been judged scientifically suitable for publication and will be formally accepted for publication once it meets all outstanding technical requirements.

Kind regards,

Muhammad Adrish

Academic Editor

PLOS ONE

Additional Editor Comments (optional):

All queries have been answered

Reviewers' comments:

Reviewer's Responses to Questions

**Comments to the Author**

1. If the authors have adequately addressed your comments raised in a previous round of review and you feel that this manuscript is now acceptable for publication, you may indicate that here to bypass the “Comments to the Author” section, enter your conflict of interest statement in the “Confidential to Editor” section, and submit your "Accept" recommendation.

Reviewer #1: All comments have been addressed

Reviewer #2: All comments have been addressed

2. Is the manuscript technically sound, and do the data support the conclusions?

Reviewer #1: Yes

Reviewer #2: Yes

3. Has the statistical analysis been performed appropriately and rigorously? 

Reviewer #1: Yes

Reviewer #2: Yes

4. Have the authors made all data underlying the findings in their manuscript fully available?

Reviewer #1: Yes

Reviewer #2: Yes

5. Is the manuscript presented in an intelligible fashion and written in standard English?

Reviewer #1: Yes

Reviewer #2: Yes

6. Review Comments to the Author

Reviewer #1: I have carefully re-read the revised manuscript "Role of ivermectin in the prevention of SARS-CoV-2 infection among healthcare workers in India: A matched case-control study". The authors have satisfactorily replied to the previous issues.

Reviewer #2: Dear Author ,

I read the revised manuscript and I 'm in agreement about the response to my comments

7. PLOS authors have the option to publish the peer review history of their article (what does this mean?). If published, this will include your full peer review and any attached files.

Reviewer #1: No

Reviewer #2: No

---

## [Editor Report · Acceptance letter]

5 Feb 2021

PONE-D-20-38031R1 

Role of ivermectin in the prevention of SARS-CoV-2 infection among healthcare workers in India: A matched case-control study 

Dear Dr. Batmanabane:

I'm pleased to inform you that your manuscript has been deemed suitable for publication in PLOS ONE. Congratulations! Your manuscript is now with our production department. 

Kind regards, 

on behalf of

Dr. Muhammad Adrish 

Academic Editor

PLOS ONE